# Entanglement evolution after a global quench across a conformal defect

**Luca Capizzi[1] and Viktor Eisler[2]**

**1** SISSA and INFN Sezione di Trieste, via Bonomea 265, I-34136 Trieste, Italy
**2** Institute of Theoretical and Computational Physics, Graz University of Technology, Petersgasse 16, A-8010 Graz, Austria

## Abstract

We study the evolution of entanglement after a global quench in a one-dimensional quantum system with a localized impurity. For systems described by a conformal field theory, the entanglement entropy between the two regions separated by the defect grows linearly in time. Introducing the notion of boundary twist fields, we show how the slope of this growth can be related to the effective central charge that emerges in the study of ground-state entropy in the presence of the defect. On the other hand, we also consider a particular lattice realization of the quench in a free-fermion chain with a conformal defect. Starting from a gapped initial state, we obtain the slope via a quasiparticle ansatz and observe small discrepancies between the field theory and lattice results, which persist even in the limit of a vanishing gap.

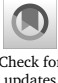

# 1 Introduction

The characterization of entanglement in quantum many-body systems has become a vast field of research [1–3]. A key result is the discovery of an area law for the entanglement entropy in ground states of local Hamiltonians [4], and its logarithmic violation in critical one-dimensional systems [5], whose features are understood via conformal field theory (CFT). In sharp contrast to the low amount of entanglement observed in the ground-state scenario, the entropy typically shows a rapid growth in the context of non-equilibrium dynamics [6–8]. In particular, it has been shown that if an initial state is suddenly perturbed by the change of a global parameter in the Hamiltonian, a protocol known as global quench [9–11], the entanglement entropy starts to grow linearly in time. This is followed by a saturation to a volume-law behaviour, suggesting the thermalization of the subsystem.

The physical interpretation of the above mechanism relies on the presence of pairs of quasiparticles, generated after the quench protocol and spreading ballistically across the system. On the level of lattice models, this is indeed realized in integrable systems, where the extensive amount of conservation laws [9] gives rise to quasiparticles with infinite lifetime. One can then identify the asymptotic state of the subsystem with a generalized Gibbs ensemble, entirely determined by the occupations of the quasiparticles, fixed solely by the initial state [12–18]. Moreover, one can argue that the very same occupations and corresponding entropy densities characterize also the dynamics, and together with the quasiparticle velocitites they completely fix the slope of the linear entropy growth [19].

The description above is known under the name of *quasiparticle picture*. While originally formulated for translational invariant systems, in recent years there has been much work devoted to generalize the setting towards inhomogeneous initial states, see e.g. the recent review [20]. In contrast, it is much less clear what happens if the translational invariance is broken by the time evolution operator itself, the simplest example being the presence of a localized defect. On one hand, the integrability of the model can be either preserved [21–23] or destroyed [24–26] by the presence of such defects, making their analytical treatment difficult. On the other hand, a class of free-fermion chains with impurities have been investigated [27–34], showing nontrivial features in their entanglement dynamics and various physical observables, depending on the details of the defect.

In this work we are interested in the characterization of entanglement growth after a global quench, in a system of two critical half-chains that are coupled together via a conformal defect. We first discuss systems described by a CFT with a conformal interface. Introducing the concept of boundary twist fields and exploiting the power of conformal symmetry, we provide a general prediction for the entropy growth between the two halves

$$S(t) = \frac{\pi c_{\text{eff}} t}{3\beta} \,, \tag{1}$$

a relation which was already derived in Ref. [35] with a different approach. Here $\beta$ is an inverse temperature characterizing the initial state, and $c_{\text{eff}} \in [0, c]$ is the so-called *effective central charge*, a parameter which depends on the transmission properties of the interface. The crucial result is that $c_{\text{eff}}$ is precisely the same parameter that appears in the ground-state entanglement scaling

$$S = \frac{c_{\text{eff}}}{6} \log \frac{L}{\epsilon} \,, \tag{2}$$

in terms of the half-chain length $L$ and cutoff $\epsilon$. In particular, $c_{\text{eff}} = 0$ if the halves are decoupled, while $c_{\text{eff}} = c$ if the system is homogeneous, with its explicit transmission dependence known only for the free boson and fermion CFT [36, 37]. Similar results were obtained also for the non-relativistic free Fermi gas [38, 39].

On the lattice level, the ground-state entropy scaling (2) has first been investigated numerically for the free-fermion and transverse Ising chains [40–42], with the analytic solution for $c_{\text{eff}}$ found in [43, 44]. Furthermore, it has been demonstrated that, for a particular lattice realization of the conformal defect, the very same $c_{\text{eff}}$ appears in the entropy dynamics after a local quench [28]. Here we shall extend these studies to the global quench from a gapped initial state and show that, while the analogous relation (1) is exact at the CFT level, it holds only approximately for the free-fermion chain. We analyze the small discrepancies between the CFT and lattice results for the slope, and conclude that they arise due to the imperfect thermal description of the initial occupation. Our results are obtained by extending the quasiparticle picture to the evolution across the defect via a proper description of the post-quench occupations. We also consider finite-size effects and discuss the special pattern of entanglement revivals [45] found for large times due to reflections.

We structure this manuscript as follows: in section 2 we give a CFT description of the global quench, exploiting the relation between Rényi entropies and boundary twist fields. We then move to the lattice part in sec. 3 and, after introducing the model, we compute the entropy growth within the quasiparticle picture. We draw our conclusions and provide an outlook in sec. 4, leaving some technical material in three appendices.

## 2 CFT results

In this section we introduce the CFT approach for the global quench in the presence of a defect. We adapt the formalism of Ref. [11] for the dynamics after the global quench, taking into account the presence of the defect. The calculation of the entanglement entropy is based on the replica trick, giving a field theoretic description of the $n$-th Rényi entropy. In particular, we compute a partition function of a $n$-sheeted Riemann surface [5], with a branch-cut starting at the interface and extending along the half-system. This approach requires the expectation value of the twist fields inserted in a bounded geometry, which is directly related to the dynamics of the Rényi entropy. We provide a general derivation of the linear growth of the entanglement entropy which does not refer to any specific model, and it is based on conformal symmetry only. Before proceeding with the calculations, it is worth to emphasize the main assumptions behind this approach.

- The post-quench Hamiltonian is described by the same CFT on both sides of the defect, thus every excitation propagates at the same speed (set to 1).

- The interface is scale-invariant, thus no dependence on the incoming momenta is present for the scattering properties across the interface (as explained in Ref. [46]).

- The initial state is a regularized boundary state $|\Psi_0\rangle \sim \exp\left(-\frac{\beta}{4}H\right)|B\rangle$, which is a short-range entangled state sharing features with a thermal state at inverse temperature $\beta$ [11].

### 2.1 Bulk and boundary twist field

Here we introduce and discuss the properties of the twist fields in a defect geometry, stressing the distinction between bulk and boundary fields. First, we briefly review the standard definition of these fields in the absence of interfaces and their connection with entanglement entropy [5]. We consider a 1+1D quantum field theory replicated $n$ times, and a pair of twist fields $\mathcal{T}(x), \tilde{\mathcal{T}}(x)$, which introduce a branch-cut along the half-line $[x, \infty)$, connecting the $k$-th and the $k \pm 1$-th replica, respectively. A precise definition can be provided via commutation

relations between the twist fields and the local fields. In particular, one requires that [47,48]

$$\mathcal{T}(x)\mathcal{O}_k(y) = \begin{cases} \mathcal{O}_{k+1}(y)\mathcal{T}(x), & x < y, \\ \mathcal{O}_k(y)\mathcal{T}(x), & \text{otherwise}, \end{cases} \qquad \tilde{\mathcal{T}}(x)\mathcal{O}_k(y) = \begin{cases} \mathcal{O}_{k-1}(y)\tilde{\mathcal{T}}(x), & x < y, \\ \mathcal{O}_k(y)\tilde{\mathcal{T}}(x), & \text{otherwise}, \end{cases} \tag{3}$$

for any local operator $\mathcal{O}_k$ inserted at a point of the $k$-th replica (here $k = 1, \dots, n$ is a replica index and its values are identified up to $k = k + n$). In a CFT, the twist fields are primary fields with scaling dimension $\Delta$ given by [5]

$$\Delta = \frac{c}{12}\left(n - \frac{1}{n}\right), \tag{4}$$

and $c$ is the central charge of the theory. Their two-point function in the (homogeneous) ground state is thus fixed by scaling symmetry only [49], and reads

$$\langle \mathcal{T}(x_1)\tilde{\mathcal{T}}(x_2)\rangle = \frac{1}{|x_1 - x_2|^{2\Delta}}. \tag{5}$$

The connection between the twist fields and entanglement arises as follows. One considers the ground state $|\Psi_0\rangle$ of a CFT, and constructs the reduced density matrix $\rho_A$ associated to a subsystem $A$ [2]. For an interval $A = [0, \ell]$, one can show that for any integer $n \geq 1$ [5]

$$\text{Tr}(\rho_A^n) = \epsilon^{2\Delta}\langle \mathcal{T}(0)\tilde{\mathcal{T}}(\ell)\rangle, \tag{6}$$

with $\epsilon$ being a UV-cutoff. Putting these informations together, one can eventually express the $n$-th Rényi entropy of $A$ as [5]

$$S_n = \frac{1}{1-n}\log\text{Tr}(\rho_A^n) = \frac{c}{6}\left(1 + \frac{1}{n}\right)\log\frac{\ell}{\epsilon}, \tag{7}$$

which gives the celebrated relation [50]

$$S \equiv S_1 = \frac{c}{3}\log\frac{\ell}{\epsilon}, \tag{8}$$

for the von Neumann entropy, once the analytical continuation $n \to 1$ is taken.

While the result (7) is specific to the translational invariant ground state, the definition (3) and the (bulk) scaling dimension (4) are general. However, there is an important caveat that arises for a bounded geometry, say an interval $[0, L]$. The value of the scaling dimension (3) refers only to a twist field $\mathcal{T}(x)$ inserted at a bulk point $(0 < x < L)$, as it is well known that boundary effects do not affect bulk dimensions [49]. Nevertheless, this is no longer true if $x$ is a boundary point $(x = 0, L)$. In that case, the dimension of the twist field, which is now a boundary field, takes in general another value, dubbed as boundary scaling dimension. Indeed, if we consider the subsystem $A = [0, \ell]$, one finds for $\ell \ll L$ [51]

$$S_n \simeq \frac{1}{1-n}\log\langle \mathcal{T}(0)\tilde{\mathcal{T}}(\ell)\rangle = \frac{c}{12}\left(1 + \frac{1}{n}\right)\log\frac{\ell}{\epsilon} + \dots, \tag{9}$$

a relation which suggests immediately that the dimension of the boundary twist field $\mathcal{T}(0)$ vanishes. This is expected on physical grounds, since the only contribution to the entanglement is given by the correlations localized around the point $x = \ell$.

So far we have considered a single theory, with or without boundaries, and we have discussed the entanglement of a single interval. Now, we generalize the construction above for a system made by two CFTs joined together through an interface. The two CFTs are denoted by $\text{CFT}_1$ and $\text{CFT}_2$, assumed here to be two copies of the same theory, and they extend over

the regions $x \in [0, L]$ and $[-L, 0]$ respectively. An interface is inserted at $x = 0$, which is a defect line extended over the Euclidean time, and gives a field theoretical representation of the impurity (see Refs. [36, 37]). We also need to specify the boundary conditions at $x = \pm L$. Here, we only assume that they do not mix the two theories, so that their precise features are essentially irrelevant for the entanglement between the two halves. We call *unfolded picture* the geometry described so far. Via the so-called folding procedure, it is possible to describe the same system as the theory $\text{CFT}_1 \otimes \text{CFT}_2$ extended over the region $x \in [0, L]$, a description which takes the name of *folded picture* (see Ref. [52] for further details). Comparing the two pictures, every point $x \in [0, L]$ in the folded geometry corresponds to a pair of points $x, -x$ (associated to $\text{CFT}_1$ and $\text{CFT}_2$ respectively) in the unfolded one. Moreover, the interface is now described via the boundary condition (BC) at $x = 0$, denoted by $b$, and similarly the BC at $x = L$ in the folded picture, denoted by $b'$. Note that the latter is just the product of the BCs at $x = \pm L$ in the unfolded picture.

The last ingredient we need to establish the correspondence among the two pictures is the characterization of the branch cuts in the folded geometry. The idea is to introduce a pair of twist fields $\mathcal{T}^j, \tilde{\mathcal{T}}^j$ ($j = 1, 2$) which act nontrivially on the degrees of freedom of $\text{CFT}_1$ or $\text{CFT}_2$ only, and use them as the building blocks for the entanglement measures among the two theories. We propose a definition, which is nothing but a generalization of Eq. (3), as follows

$$
\begin{aligned}
\mathcal{T}^j(x)\mathcal{O}_{j',k}(y) &= \begin{cases} \mathcal{O}_{j',k+1}(y)\mathcal{T}^j(x), & x < y, \quad j = j', \\ \mathcal{O}_{j',k}(y)\mathcal{T}^j(x), & \text{otherwise}, \end{cases} \\
\tilde{\mathcal{T}}^j(x)\mathcal{O}_{j',k}(y) &= \begin{cases} \mathcal{O}_{j',k-1}(y)\tilde{\mathcal{T}}^j(x), & x < y, \quad j = j', \\ \mathcal{O}_{j',k}(y)\tilde{\mathcal{T}}^j(x), & \text{otherwise}, \end{cases}
\end{aligned}
\tag{10}
$$

with $\mathcal{O}_{j,k}(y)$ being local operators of $(\text{CFT}_1 \otimes \text{CFT}_2)^{\otimes n}$ with species index $j = 1, 2$ and replica index $k = 1, \ldots, n$. As an illustrative example, the insertion of $\mathcal{T}^1(x_1)\tilde{\mathcal{T}}^1(x_2)$ at two points $0 < x_1 < x_2 < L$ of the folded geometry, corresponds to the insertion of $\mathcal{T}(x_1)\tilde{\mathcal{T}}(x_2)$ in the unfolded picture, which is a branch cut along a segment of $\text{CFT}_1$. Similarly, $\mathcal{T}^2(x_1)\tilde{\mathcal{T}}^2(x_2)$ introduces a branch cut on $[-x_2, -x_1]$ in the unfolded picture. We summarize the construction above in Fig. 1, which provides a pictorial representation of the insertion of the branch cuts in the folded/unfolded pictures.

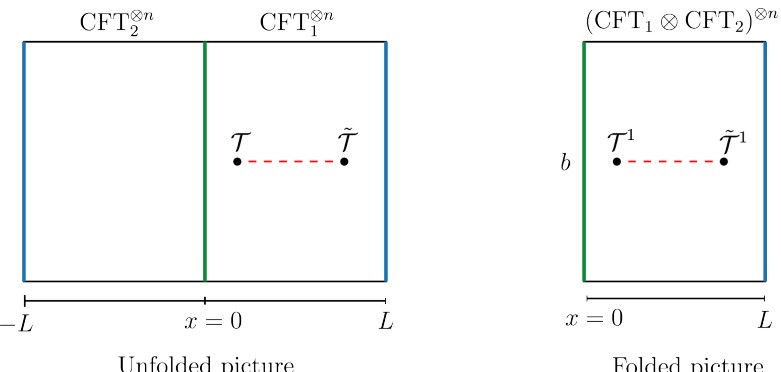

Figure 1: Representation of a system made by two CFTs joined through an interface in the unfolded (left) and folded (right) picture. The interface and the open boundaries are shown by green and blue lines, and are associated to boundary conditions of type $b$ and $b'$, respectively.

As long as the twist fields $\mathcal{T}^j$ are inserted at bulk points, their (bulk) scaling dimension is simply given by Eq. (4). In contrast, the dimension of $\mathcal{T}^j(L)$ is expected to be zero, since the

two theories are not coupled at $x = L$ and one can use the results available for a single theory. Moreover, we argue that the scaling dimension of $\mathcal{T}^j(0)$ is nontrivial, and strongly depends on the boundary condition $b$ (and $n$), representing the interface: we denote it by $\Delta_b$. We thus compute the Rényi entropy among the two halves as

$$S_n = \frac{1}{1-n} \log\left(\epsilon^{\Delta_b} \langle \mathcal{T}^1(0)\tilde{\mathcal{T}}^1(L)\rangle\right) = \frac{\Delta_b}{n-1}\log\frac{L}{\epsilon} + \dots, \tag{11}$$

where the last equality follows from scaling arguments only, and subleading terms of order $O(1)$ have been neglected. More precisely, if one applies a scale transformation, say $L \to L\lambda$, then the geometry becomes $[0, \lambda L]$ and the twist fields are inserted at the boundary of this new geometry ($x = 0, \lambda L$). An additional scale factor $\lambda^{\Delta_b}$ appears due to the scaling transformation of the twist fields. Taking for example $\lambda = 1/L$ for the scaling parameter, one completely fixes the $L$-dependence of the two-point function which gives directly Eq. (11).

Eventually, one concludes that the entropy grows as the logarithm of the size, with an interface dependent prefactor, a result which was firstly obtained for the free boson in [36] and for Ising/free fermions in [37]. In particular, in the limit $n \to 1$ one has

$$S = \frac{c_{\text{eff}}}{6}\log\frac{L}{\epsilon}, \tag{12}$$

where $c_{\text{eff}}$ is a parameter dubbed as *effective central charge*. Comparing the last expression with our formula (11), we establish the equivalence

$$\frac{c_{\text{eff}}}{6} = \lim_{n\to 1}\frac{\Delta_b}{n-1}. \tag{13}$$

It tells us that the origin of the nontrivial scaling of entanglement for systems with conformal defects can be traced back to a nontrivial boundary scaling dimension of the twist field $\mathcal{T}^j$.

## 2.2 Global quench

We continue with the study of a global quench for a CFT in the presence of a defect, generalizing the approach of [11] valid without the defect. We mention that a similar method to tackle the problem has been already employed in [35]. Here, we consider instead a novel technique based on the use of the boundary twist fields.

We work in the folded picture, introduced in the previous subsection, and consider the theory $\text{CFT}_1 \otimes \text{CFT}_2$ extended over the spacial region $[0, L]$. We take the initial state

$$|\Psi_0\rangle \sim \exp\left(-\frac{\beta}{4}H\right)|b'\rangle, \tag{14}$$

with $|b'\rangle$ being a boundary state associated to the BC $b'$, $H$ the post-quench Hamiltonian and $\beta$ a parameter representing the inverse temperature of the initial state. Within this choice, the initial state is short-range entangled, having a finite correlation length of order $\beta$. Since $|\Psi_0\rangle$ is not an eigenvector of $H$, the unitary dynamics induced by the Hamiltonian is nontrivial. We consider time scales much shorter than the size $L$, $t \ll L$, so that finite-size effects are negligible. We express the Rényi entropy between the two halves as

$$S_n(t) = \frac{1}{1-n}\log\left(^{\otimes n}\langle\Psi_0| e^{iHt}\mathcal{T}^1(0)\tilde{\mathcal{T}}^1(L)e^{-iHt}|\Psi_0\rangle^{\otimes n}\right), \tag{15}$$

with $|\Psi_0\rangle^{\otimes n} = |\Psi_0\rangle \otimes \cdots \otimes |\Psi_0\rangle$ being a state of the $n$-replica theory $(\text{CFT}_1 \otimes \text{CFT}_2)^{\otimes n}$. Since $\tilde{\mathcal{T}}^1(L)$ has dimension 0, we simply drop it from Eq. (15), a procedure which is physically motivated by the fact that its insertion should not affect the entanglement among the two

halves (see [47] for futher details). Taking into account the correct normalization of the state, we finally end up with

$$S_n(t) = \frac{1}{1-n} \log \frac{{}^{\otimes n}\langle b'| e^{-H(\beta/4 - it)} \mathcal{T}^1(0) e^{-H(\beta/4 + it)} |b'\rangle^{\otimes n}}{{}^{\otimes n}\langle b'| e^{-H\beta/2} |b'\rangle^{\otimes n}}. \tag{16}$$

We now proceed further with the evaluation of the expectation value appearing in Eq. (16). We work in the Euclidean theory, described as the bounded geometry (strip)

$$\mathrm{Re}(z) \in [0, +\infty), \quad \mathrm{Im}(z) \in [-\beta/4, \beta/4], \tag{17}$$

obtained in the limit $L \to \infty$. We insert the twist field $\mathcal{T}^1$ at the position $z = i\tau$ of the strip, with $\tau$ being the Wick rotated time

$$\tau = it. \tag{18}$$

To compute $\langle \mathcal{T}^1(z = i\tau)\rangle$, we map the strip (17) onto the upper half-plane $\mathrm{Im}(w) \geq 0$ through a sequence of transformations $z \to \zeta \to w$ defined by

$$\zeta(z) = \sin \frac{2\pi i z}{\beta}, \quad w(\zeta) = \frac{1+\zeta}{2(1-\zeta)}. \tag{19}$$

Via this change of variables, the corners at $z = \pm i\beta/4$ are mapped onto the points $w = 0, \infty$ respectively. In this new geometry a change of boundary conditions, from type $b'$ to type $b$, appears at $w = 0$ along the real line $\mathrm{Im}(w) = 0$. In Fig. 2 we represent the insertion of the twist field both in the strip geometry ($z$ variable) and the upper half-plane ($w$ variable).

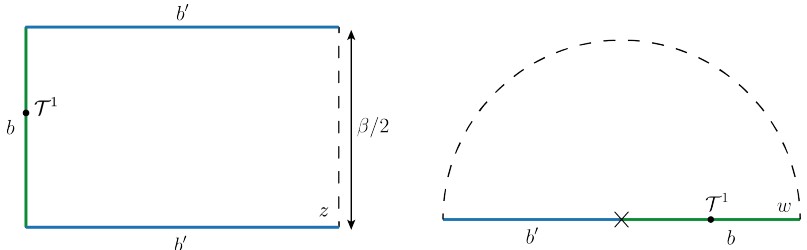

Figure 2: Euclidean boundary geometry describing the global quench in the $z$ (left) and $w$ (right) variable. Green/blue colors denote boundary conditions of type $b/b'$, respectively.

At this point, we observe explicitly that the depicted geometry (upper half-plane) shows a scale-symmetry

$$w \to \lambda w, \quad \lambda > 0. \tag{20}$$

Indeed, the boundary conditions $b', b$ are scale-invariant and the point $w = 0$, representing the change of BCs, is kept fixed. We use this property to infer the value of the one-point function along the boundary of type $b$ ($\mathrm{Im}(w) = 0, w > 0$) which is

$$\langle \mathcal{T}^1(w)\rangle \propto \frac{1}{|w|^{\Delta_b}}, \tag{21}$$

up to a $w$-independent proportionality constant not further specified.

We now go back to the original geometry, employing the transformation law of the twist field $\mathcal{T}^1$, which is a primary operator. We firstly apply $w \to \zeta$, reaching to

$$\langle \mathcal{T}^1(\zeta)\rangle = \left|\frac{dw}{d\zeta}\right|^{\Delta_b} \langle \mathcal{T}^1(w)\rangle \propto \left|\frac{1}{(1-\zeta)^2}\right|^{\Delta_b} \left|\frac{1-\zeta}{1+\zeta}\right|^{\Delta_b} = \left|\frac{1}{1-\zeta^2}\right|^{\Delta_b}, \tag{22}$$

and similarly from the map $\zeta \to z$ we get

$$\langle \mathcal{T}^1(z) \rangle = \left| \frac{d\zeta}{dz} \right|^{\Delta_b} \langle \mathcal{T}^1(\zeta) \rangle \propto \left| \frac{4\pi}{\beta \cos(2\pi i z/\beta)} \right|^{\Delta_b} . \tag{23}$$

Performing the Wick rotation $\tau = it$, for large values $t/\beta$ we obtain

$$\langle \mathcal{T}(z = i\tau) \rangle \sim \left| \frac{8\pi}{\beta} e^{-2\pi t/\beta} \right|^{\Delta_b} , \tag{24}$$

which leads directly to

$$S_n(t) - S_n(0) = \frac{1}{1-n} \log \frac{\langle \mathcal{T}^1(z = i\tau) \rangle}{\langle \mathcal{T}^1(z = 0) \rangle} \simeq \frac{\Delta_b}{n-1} \frac{2\pi t}{\beta} , \tag{25}$$

the main result of this subsection. One can further take the analytical continuation $n \to 1$ from Eq. (25) to extract the entanglement entropy, and, using Eq. (13), one gets

$$S_1(t) - S_1(0) \simeq \frac{\pi c_{\text{eff}} t}{3\beta} . \tag{26}$$

We conclude with some remarks regarding the derivation provided above. For instance, even if we did not attempt to calculate explicitly the scaling dimension $\Delta_b$ of the boundary twist field, we got a precise relation between equilibrium (11) and non-equilibrium entropy (25) based on conformal invariance only. Moreover, the same derivation can be applied to the evolution of any boundary (primary) operator, and it is not specific of the twist field. We finally stress again that the linear growth in (25) is expected to be valid for $\beta \ll t \ll L$, so that the short-time effects, as well as the long-time recurrences, are neglected.

# 3 Lattice results

In the following we consider a lattice realization of the global quench. Our main goal is to derive, analogously to the CFT calculations, a formula that relates the entropy growth in the presence of a defect to that in a homogeneous chain. We show that, in contrast to CFT results that predict the appearance of the same proportionality factor between the Rényi entropies as in the ground state, this is not true in general for the lattice. We first introduce our model and summarize the known results about its ground-state entanglement, followed by a discussion of the quench setup.

## 3.1 Model and setup

We consider a fermionic hopping chain of length $2L$ with Hamiltonian

$$\hat{H} = \sum_{m,n=-L+1}^{L} H'_{m,n} c^\dagger_m c_n , \tag{27}$$

where $c^\dagger_m$ and $c_m$ are fermionic creation/annihilation operators satisfying anticommutation relations $\{c^\dagger_m, c_n\} = \delta_{m,n}$. We focus on a particular form of a defect located in the center of the chain, with the nonzero elements of the hopping matrix given by

$$H'_{m,m+1} = H'_{m+1,m} = \begin{cases} -1/2, & m \neq 0, \\ -\lambda/2, & m = 0, \end{cases} \qquad H'_{0,0} = -H'_{1,1} = \sqrt{1-\lambda^2}/2. \tag{28}$$

The defect of strength $\lambda$ thus consists of a single modified hopping amplitude as well as local chemical potentials across sites 0 and 1. Throughout the manuscript, we use the prime notation to refer to various quantities of the defect problem, whereas the same symbols without prime shall refer to the homogeneous ($\lambda = 1$) case.

The defect problem defined by the hopping matrix $H'$ was studied before in [28] and was shown to have a particularly simple relation to the homogeneous case $H$, where the single-particle eigenvalues and eigenvectors are given by

$$\phi_k(m) = \sqrt{\frac{2}{2L+1}} \sin \frac{\pi k(m+L)}{2L+1}, \qquad \omega_k = -\cos \frac{\pi k}{2L+1}, \tag{29}$$

for $k = 1, \ldots, 2L$. Indeed, the corresponding quantities for the defect are simply related via [28]

$$\phi'_k(m) = \begin{cases} \alpha_k \phi_k(m), & m \leq 0, \\ \beta_k \phi_k(m), & m \geq 1, \end{cases} \qquad \omega'_k = \omega_k, \tag{30}$$

where the scaling factors read

$$\alpha_k^2 = 1 + (-1)^k \sqrt{1-\lambda^2}, \qquad \beta_k^2 = 1 - (-1)^k \sqrt{1-\lambda^2}. \tag{31}$$

In other words, the eigenvectors of $H'$ are only rescaled by a different factor on the left/right hand side of the defect, while the spectrum remains unchanged.

The relation (30) has important consequences on the transmission properties of the defect, which are easier to discuss in the thermodynamic limit $L \to \infty$ of the chain. One then writes the following plane-wave ansatz for the eigenvectors $\bar{\phi}_q(m)$ and $\phi_q(m)$ on the left/right hand side of the defect

$$\bar{\phi}_q(m) = \begin{cases} A_1 e^{iqm} + B_1 e^{-iq(m-1)}, & q > 0, \\ A_2 e^{iqm}, & q < 0, \end{cases} \qquad \phi_q(m) = \begin{cases} A_2 e^{iqm}, & q > 0, \\ A_1 e^{iqm} + B_1 e^{-iq(m-1)}, & q < 0. \end{cases} \tag{32}$$

For a right-moving ($q > 0$) particle, $\bar{\phi}_q(m)$ is just the superposition of an incoming and reflected wave, whereas $\phi_q(m)$ corresponds to the transmitted wave component (and vice-versa for $q < 0$). The coefficients are fixed by matching the solutions at the defect via

$$\lambda \bar{\phi}_q(0) + \sqrt{1-\lambda^2}\,\phi_q(1) = \phi_q(0), \qquad \lambda \phi_q(1) - \sqrt{1-\lambda^2}\,\bar{\phi}_q(0) = \bar{\phi}_q(1), \tag{33}$$

and yield the following transmission and reflection amplitudes

$$s(q) = \frac{A_2}{A_1} = \lambda, \qquad r(q) = \frac{B_1}{A_1} = -\text{sign}(q)\sqrt{1-\lambda^2}. \tag{34}$$

The transmission $T = |s(q)|^2 = \lambda^2$ and reflection $R = |r(q)|^2 = 1 - T$ coefficients are thus independent of the wavenumber, hence it has been dubbed as a *conformal defect* [28]. Note, however, that there is a nontrivial sign factor in the reflection amplitude $r(q)$, required by the orthonormality of the basis, which also yields $|A_1|^2 = 1/2\pi$ for the incoming amplitude.

The simple structure (30) of the eigenvectors has also important implications on the ground-state entanglement properties between the two halves $A = [-L+1, 0]$ and $B = [1, L]$ of the chain. These follow from the fermionic reduced correlation matrix $C'_A$ with elements [53]

$$C'_{m,n} = \langle c_m^\dagger c_n \rangle = \sum_{k=1}^{L} \phi'_k(m)\phi'_k(n), \qquad m, n \in A. \tag{35}$$

In particular, the entanglement entropy of the reduced density matrix $\rho_A = \text{Tr}_B \rho$ is obtained as

$$S = \sum_{l=1}^{L} s(\zeta_l'), \qquad s(x) = -x \ln x - (1-x) \ln(1-x), \tag{36}$$

where $\zeta_l'$ are the eigenvalues of $C_A'$. Using (30) for the matrix elements in (35), one finds the exact relation for the ground state of the conformal defect

$$C_A'(1 - C_A') = \lambda^2 C_A(1 - C_A), \tag{37}$$

which yields an analogous relation between the eigenvalues. Equivalently, one could rewrite (37) in terms of the single-particle entanglement spectrum

$$\zeta_l' = \frac{1}{e^{\varepsilon_l'} + 1}, \tag{38}$$

which then yields the relation

$$\cosh \frac{\varepsilon_l'}{2} = \frac{1}{\lambda} \cosh \frac{\varepsilon_l}{2}. \tag{39}$$

The latter relation can then be used to find the scaling of the entanglement entropy which reads

$$S = \kappa(\lambda) \ln L, \tag{40}$$

where the prefactor of the leading logarithmic term is obtained as [43]

$$\kappa(s) = -\frac{1}{\pi^2} \left\{ \left[ (1+s) \ln(1+s) + (1-s) \ln(1-s) \right] \ln s + (1+s) \text{Li}_2(-s) + (1-s) \text{Li}_2(s) \right\}, \tag{41}$$

in terms of the transmission amplitude $s = \lambda$. Note that the prefactor could also be written as $\kappa(\lambda) = c_{\text{eff}}/6$ via an effective central charge, which varies smoothly between one and zero, corresponding to the purely transmissive ($\lambda = 1$) and reflective cases ($\lambda = 0$). In fact, (39) can also be used to extract the scaling of the Rényi entropies of integer index $n > 1$

$$S_n = \sum_{l=1}^{L} s_n(\zeta_l'), \qquad s_n(x) = \frac{1}{1-n} \ln \left[ x^n + (1-x)^n \right]. \tag{42}$$

The result is completely analogous to (40), with the corresponding prefactor given by [44]

$$\kappa_n(\lambda) = \frac{1}{\pi^2} \frac{1}{n-1} \sum_{p=-(n-1)/2}^{(n-1)/2} \arcsin^2 \left[ \lambda \sin\left( \frac{p\pi}{n} \right) \right], \tag{43}$$

where the sum over $p$ runs over half-integer/integer values for $n$ even/odd.

We now turn our attention to the quench setup. In order to mimic the global quench scenario of the CFT setting, we choose a gapped initial Hamiltonian $\hat{H}_0$ and prepare the chain in its ground state. The simplest way to open a gap is to add a staggered chemical potential to the Hamiltonian (27) of the conformal defect

$$\hat{H}_0 = \hat{H} + \mu \sum_{n=-L+1}^{L} (-1)^n c_n^\dagger c_n. \tag{44}$$

The quench then consists of switching off the potential at time $t = 0$ and time evolve the initial state with $\hat{H}$. The size of the gap in $\hat{H}_0$ is controlled by the mass parameter $\mu$, and in the limit $\mu \to \infty$ one obtains the Néel state, where each odd lattice site is occupied by a particle with the even sites being empty. In fact, the very same initial Hamiltonian (44) was considered in Ref. [35] to test the CFT predictions for the global quench numerically, finding a good agreement. Here we revisit this problem more carefully, providing analytic results for the quench on the lattice that show some deviations from the CFT result. We start by discussing the homogeneous problem first.

## 3.2 Quasiparticle ansatz for homogeneous case

The entropy growth after a global quench in an integrable translational invariant chain can be understood in terms of a quasiparticle picture [10,19,54]. Namely, one assumes that the initial state is a source of quasiparticle pair excitations that propagate linearly under the dynamics, and carry the initial short range entanglement to large distances. In a free-fermion chain, each mode with momentum $q$ propagates with the corresponding group velocity $v_q = \sin q$, and a pair contributes to the entanglement at time $t$ if one particle is located in subsystem $A$ while the other one in $B$. The contribution of a given pair is simply the thermodynamic entropy density taken in the steady state of the dynamics, which is uniquely determined by the occupation numbers $n_q = \langle c_q^\dagger c_q \rangle$, that are constants of motion. Putting everything together, in the limit $L \to \infty$ of a semi-infinite subsystem one arrives at the expression

$$S_n(t) = t \int_{-\pi}^{\pi} \frac{dq}{2\pi} |v_q| s_n(n_q), \qquad (45)$$

where the entropy density $s_n$ was defined in (42), see also (36) for $n = 1$. The quasiparticle ansatz (45) in the half-chain geometry thus gives a purely linear growth of entanglement, $S_n(t) = \gamma_n(\mu, 1)t$, where the second argument of the slope refers to $\lambda = 1$. In order to find the slope $\gamma_n(\mu, 1)$ as a function of the quench parameter $\mu$, one has to evaluate the occupation numbers $n_q$ characterising the steady state after the quench. These are fixed by the ground state of the staggered chain, and can be found by diagonalizing $\hat{H}_0$ as shown in appendix A, with the result

$$n_q = \frac{1}{2}\left(1 + \frac{\cos q}{\sqrt{\cos^2 q + \mu^2}}\right). \qquad (46)$$

The occupation function is shown in Fig. 3 for various values of $\mu$. In the limit $\mu \to 0$ it converges towards the Fermi sea occupation, whereas for the Néel state, $\mu \to \infty$, it becomes completely flat, $n_q = 1/2$.

With the result (46) for the occupation at hand, one can now compare the quasiparticle ansatz (45) to the entropy obtained from the lattice calculation. We shall restrict ourselves to the case $n = 1$. This can be obtained from (36), using the eigenvalues $\zeta_l(t)$ of the time-evolved reduced correlation matrix $C_A(t)$. The correlations at time $t$ are given by $C(t) = U^\dagger C(0) U$, where $C(0)$ describes the correlations of the staggered ground state, and the propagator is given by $U = e^{-iHt}$. For our numerics we choose $L = 100$, and consider the short-time regime $t < L$, where the presence of the boundaries cannot yet influence the behaviour. The result is shown in Fig. 4 for various values of $\mu$. On the left, the linear growth with slopes $\gamma(\mu, 1) \equiv \gamma_1(\mu, 1)$ calculated from (45) are plotted by red solid lines, showing a very good agreement with the data. Note that we have subtracted the initial value $S(0)$ of the entropy. On the right of Fig. 4 we show the corrections to the quasiparticle ansatz by subtracting the linear piece. Interestingly, the leading correction seems to be logarithmic in time, albeit with a very small prefactor that increases for larger $\mu$.

## 3.3 Quench with a defect: Short times

We now move on to consider the quench with the defect. Motivated by the CFT results, our goal is to establish a relation between the defect and the homogeneous cases, analogous to the one (37) found for the ground state. The first step is to calculate the defect propagator $U' = e^{-iH't}$. Using (32), the matrix elements on the same side of the defect are obtained as

$$U'_{m,n} = \int_{-\pi}^{\pi} \frac{dq}{2\pi} e^{-i\omega_q t} \left[ e^{-iq(m-n)} \pm \sqrt{1-\lambda^2}\, e^{-iq(m+n-1)} \right] = U_{m,n} \pm \sqrt{1-\lambda^2}\, U_{m,1-n}, \qquad (47)$$

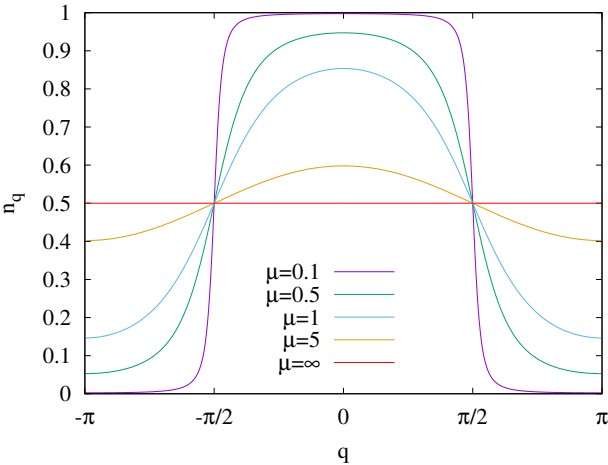

Figure 3: Occupation numbers (46) for various values of $\mu$.

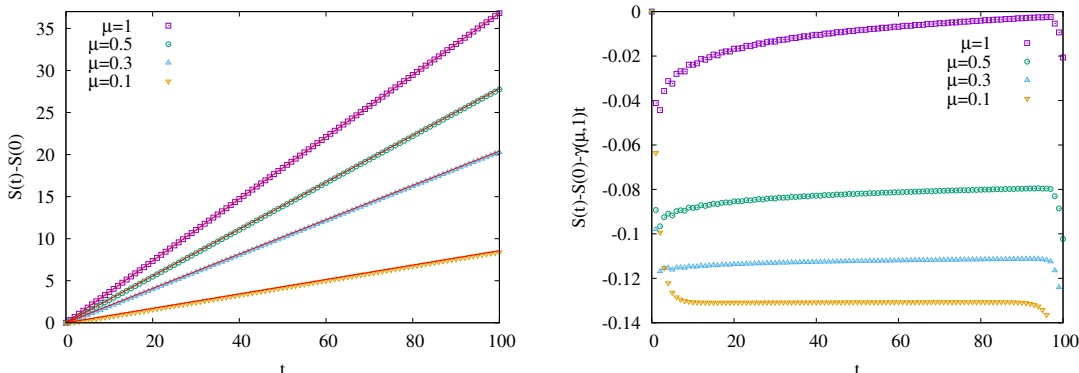

Figure 4: Left: entropy evolution for a homogeneous chain ($\lambda = 1$) with $L = 100$ after a global quench for various values of $\mu$. The red lines show the slopes calculated from (45) with the occupation (46). Right: deviation from the quasiparticle ansatz.

where the $\pm$ sign refers to the case $m, n \geq 1$ and $m, n \leq 0$, respectively. The defect propagator thus differs from the homogeneous one in the presence of a reflected component that is multiplied by the corresponding amplitude. On the other hand, for the off-diagonal terms with $m \leq 0$ and $n \geq 1$ as well as $m \geq 1$ and $n \leq 0$ one obtains

$$U'_{m,n} = \lambda \int_{-\pi}^{\pi} \frac{\mathrm{d}q}{2\pi} \, e^{-i\omega_q t} e^{-iq(m-n)} = \lambda \, U_{m,n}. \tag{48}$$

Hence, for indices on opposite sides of the defect the propagator is just multiplied by the transmission amplitude.

For the calculation of the entropy we need the time-evolved matrix $C'(t) = U'^{\dagger} C'(0) U'$, where $C'(0)$ contains the ground-state correlations of the staggered Hamiltonian (44) with a defect. In general, this is a rather complicated object of which we do not have a closed-form expression. Hence, for the purpose of this calculation we shall restrict ourselves to the Néel case ($\mu \to \infty$), where $C'(0)$ is diagonal and independent of $\lambda$. In order to establish the analogue of (37), let us first note that, due to the purity of the time-evolved state, the full correlation matrix satisfies $C'(t)(1 - C'(t)) = 0$. This yields

$$C'_A(t)(1 - C'_A(t)) = C'_{AB}(t) C'_{BA}(t), \tag{49}$$

where $C'_{AB}(t)$ is the off-diagonal part of the correlation matrix, and $C'_{BA}(t) = C'^{\dagger}_{AB}(t)$. Its matrix elements with the Néel initial state read

$$C'_{m,n}(t) = \sum_{l=-\infty}^{\infty} U'^{*}_{2l-1,m} U'_{2l-1,n}, \tag{50}$$

where $m \leq 0$ and $n \geq 1$. Using the relations (47) and (48), this leads to

$$\sum_{l \leq 0} (U^{*}_{2l-1,m} - \sqrt{1-\lambda^2}\, U^{*}_{2l-1,1-m})\lambda U_{2l-1,n} + \sum_{l \geq 1} \lambda U^{*}_{2l-1,m}(U_{2l-1,n} + \sqrt{1-\lambda^2}\, U_{2l-1,1-n}), \tag{51}$$

which, using the reflection symmetry of the propagator $U_{m,n} = U_{1-m,1-n}$, can be rewritten as

$$\lambda \sum_{l} U^{*}_{2l-1,m} U_{2l-1,n} + \lambda \sqrt{1-\lambda^2} \sum_{l \geq 1} (U^{*}_{2l-1,m} U_{2l-1,1-n} - U^{*}_{2l,m} U_{2l,1-n}). \tag{52}$$

In other words, one finds

$$C'_{AB}(t) = \lambda C_{AB}(t) + \lambda \sqrt{1-\lambda^2} \Delta C_{AB}(t), \tag{53}$$

where $\Delta C_{AB}(t)$ denotes the nonvanishing contribution from the second sum in (52).

The expression obtained in (53) is thus not quite the one we expect. Indeed, the exact analogue of the ground-state relation (37) would be reproduced via (49) only for $\Delta C_{AB}(t) \equiv 0$, which is not the case here. The interpretation of this result is as follows. The first term of (53) describes the process where one of the quasiparticles gets transmitted across the defect, picking up an amplitude $\lambda$, while its pair simply propagates in the homogeneous chain. On the other hand, the prefactor $\lambda\sqrt{1-\lambda^2}$ of the second term shows, that the corresponding contribution is carried by the partially transmitted and reflected wave. Although such a process generates some nontrivial correlations between the two halves of the chain, we argue that its contribution to the entanglement must be subleading. Indeed, the main idea of the quasiparticle picture is that the entanglement is carried by counterpropagating modes created at time $t = 0$ all along the chain. This contribution gets reduced by the transmission amplitude $s(q) = \lambda$ which is completely independent of $q$. On the other hand, however, the reflection amplitude $r(q)$ in (34) contains a $\mathrm{sign}(q)$ factor, and thus the dynamics generates a distructive interference between the partially reflected contributions of pairs emitted from opposite sides of the defect. In turn, the nonvanishing of the $\Delta C_{AB}(t)$ term is actually due to the imperfect symmetry of the initial state.

The situation for generic $\mu$ is even more complicated, as the initial correlations become more and more long-ranged as $\mu \to 0$ and the contributions from the off-diagonal blocks of $C'(0)$ become relevant as well. The corresponding relation (53) thus becomes more complicated, with additional terms appearing with prefactors that mirror the relevant scattering process. Nevertheless, similarly to our above argument, we expect that these are due to imperfect destructive interference and thus are higher order effects from the viewpoint of entanglement dynamics. Hence, we expect that the relation for the eigenvalues

$$\zeta'_l(t)(1 - \zeta'_l(t)) \simeq \lambda^2 \zeta_l(t)(1 - \zeta_l(t)), \tag{54}$$

should still hold approximately, even though the corresponding relation between the matrices $C'_A(t)$ and $C_A(t)$ does not hold, not even in a perturbative sense. The eigenvalue relation is tested in Fig. 5 for a chain of size $L = 100$, time $t = 50$ and two different values of $\mu$, with full/empty symbols corresponding to the left/right hand side of Eq. (54). One can see a very good overlap between the two quantities, with only slight shifts between the symbols, except for the spectral edges where the discrepancy becomes larger. Similar results can be obtained for arbitrary times $t < L$, with the number of nonzero data points increasing with $t$.

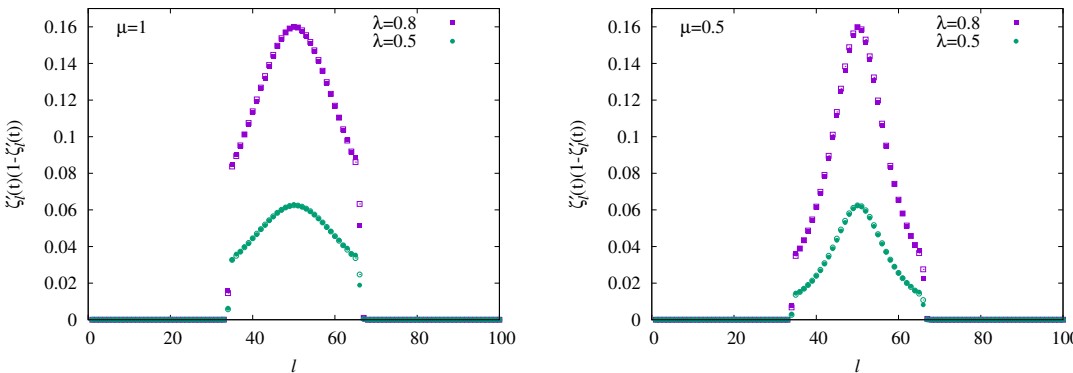

Figure 5: Numerical check of the relation (54), with the left/right hand side of the equation shown by the full/empty symbols, for $L = 100$, $t = 50$ and two different values of $\mu$.

According to the quasiparticle picture, relation (54) should actually be interpreted as a relation between the occupation functions after the quench

$$n'_q(1 - n'_q) = \lambda^2 n_q(1 - n_q). \tag{55}$$

Using the homogeneous result (46), this equation can be solved for the defect occupations as

$$n'_q = \frac{1}{2}\left(1 \pm \sqrt{1 - \lambda^2 \frac{\mu^2}{\cos^2 q + \mu^2}}\right). \tag{56}$$

Hence, the slope of the entropy growth is given by

$$\gamma_n(\mu, \lambda) = \int_{-\pi}^{\pi} \frac{dq}{2\pi} |v_q| s_n(n'_q). \tag{57}$$

The result (57) is compared against the lattice data in Fig. 6 for a small and intermediate value of the mass parameter, and various defect strengths $\lambda$. One observes a very good agreement, with the deviations from the quasiparticle result shown on the inset.

The subleading term for $\mu = 1$ is likely to be logarithmic in time, with some superimposed oscillations, whereas it seems to be given by a constant for $\mu = 0.1$. However, a closer inspection of the latter case indicates that the corrections are probably still logarithmic, albeit with a tiny prefactor, which is also supported by calculations for $L = 200$.

With the expression (57) for the slope at hand, one can now compare the result to the CFT prediction. This tells us that the slope ratio $\gamma_n(\mu, \lambda)/\gamma_n(\mu, 1)$ should be equal to the ground-state entropy ratio $\kappa_n(\lambda)/\kappa_n(1)$. Let us first consider the Néel limit, $\mu \to \infty$, where the occupation $n'_q$ in (56) becomes piecewise constant, and the integral (57) simply evaluates to

$$\gamma_n(\infty, \lambda) = \frac{2}{\pi} s_n\left(\frac{1 + \sqrt{1 - \lambda^2}}{2}\right). \tag{58}$$

Obviously, this result has a rather different analytical behaviour as compared to the ground-state prefactors (43). Remarkably, however, the ratios turn out to be very close to each other, as shown by the comparison on the left of Fig. 7 for $n = 1, 2$, although the discrepancy increases with $n$. In fact, in the limit of large mass $\mu$, one does not expect the CFT result to be exact, since one obtains contributions from the entire Brillouin zone and thus the role of the lattice dispersion enters. In contrast, in the limit $\mu \to 0$ only the modes around the Fermi level

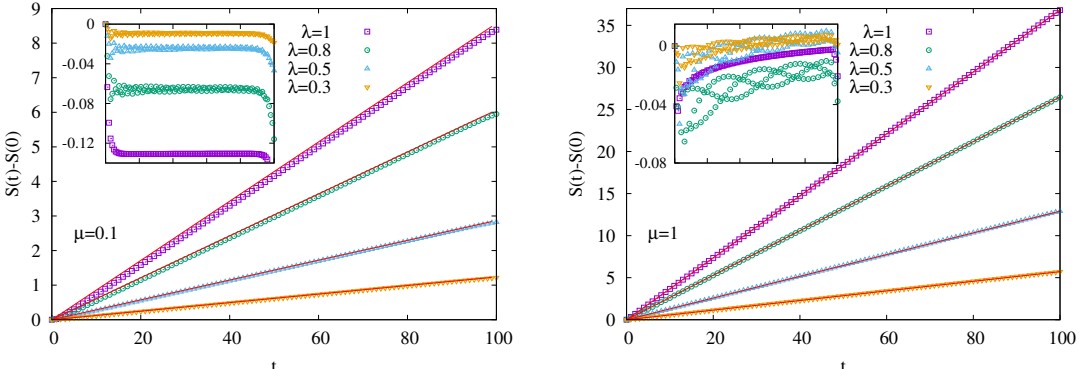

Figure 6: Entropy growth after a global quench with mass parameters $\mu = 0.1$ (left) and $\mu = 1$ (right), for various defect strengths $\lambda$ and $L = 100$. The red lines show the quasiparticle ansatz with slope (57), while the insets show the corresponding deviations.

$q = \pm\pi/2$ contribute (see Fig. 3), and one would expect the CFT description to become exact. Surprisingly, this turns out not to be the case, as demonstrated on the right of Fig. 7. Indeed, in the limit $\mu \to 0$, the slope vanishes linearly with the mass and one can find the closed analytical expression for $n \neq 1$

$$\lim_{\mu \to 0} \frac{\gamma_n(\mu, \lambda)}{\mu} = \frac{1}{n-1} \sum_{p=-(n-1)/2}^{(n-1)/2} \left[ 1 - \sqrt{1 - \lambda^2 \sin^2(\pi p/n)} \right], \tag{59}$$

as shown in appendix B. The ratio $\gamma_n(0, \lambda)/\gamma_n(0, 1)$ is thus finite, and the structure of (59) shows a close resemblance to that in (43), although the expressions in the sum are different. Nevertheless, the mismatch from the ratio $\kappa_n(\lambda)/\kappa_n(1)$ turns out to be very small again, and the curves now approach the CFT limits from the other side, as compared to the $\mu \to \infty$ case.

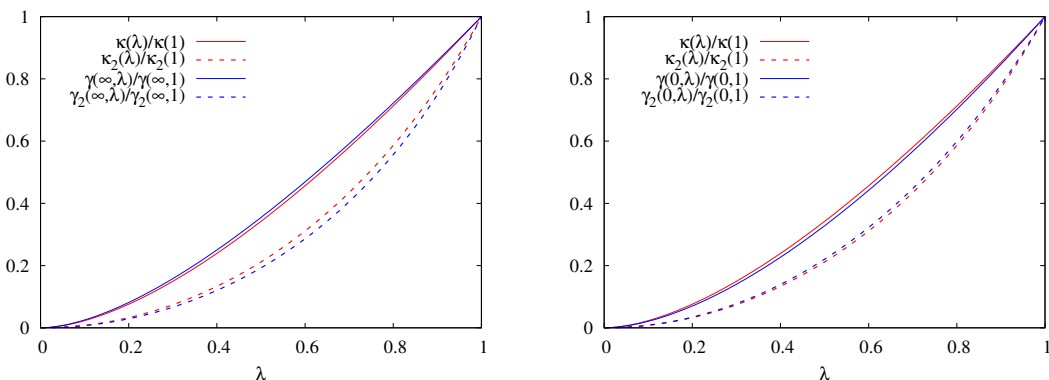

Figure 7: Comparison of the slope ratios (blue) to the CFT prediction (red) in the limits $\mu \to \infty$ (left) and $\mu \to 0$ (right), as a function of $\lambda$. The solid/dashed lines correspond to $n = 1$ and $n = 2$.

For general $\mu$, the behaviour of the discrepancy $\gamma(\mu, \lambda)/\gamma(\mu, 1) - \kappa(\lambda)/\kappa(1)$ between the slope ratios is shown for some fixed values of $\lambda$. The deviation remains rather small in the entire regime $0.1 \leq \mu \leq 5$ shown. In particular, it decreases for $\lambda \to 0$ and $\lambda \to 1$, while the maximal deviations are observed around $\lambda \simeq 0.5$, similarly to Fig. 7. Interestingly, the curves change sign at around $\mu \simeq 0.3$, although they do not intersect at the same $\mu$ as it might seem.

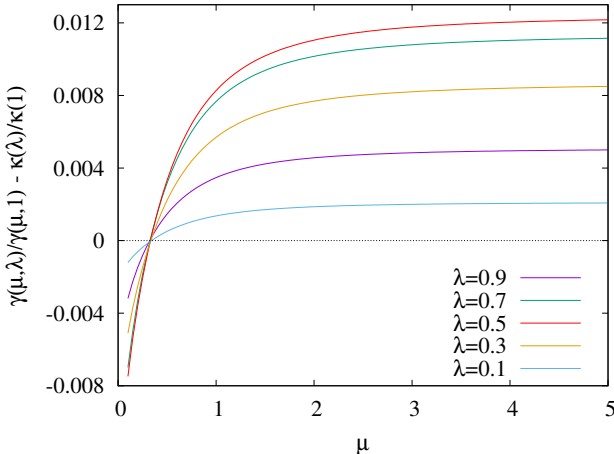

Figure 8: Deviation of the slope ratio from the CFT prediction with $n = 1$, plotted in the regime $0.1 \leq \mu \leq 5$ for various fixed values of $\lambda$.

Thus, in general, the lattice quench ratios are governed by a different function of $\lambda$ from the one in CFT, albeit with such a small discrepancy that could not have been found by fitting the data, without the analytical solution of the problem.

The reason of the discrepancy is that, even in the limit $\mu \to 0$, the occupation (46) does not correspond to a thermal distribution, which is implicitly assumed in the CFT treatment. In fact, the calculation of the slope can be generalized to this case, since the relation (55) holds for *arbitrary* initial occupations. Choosing a thermal initial state of the homogeneous chain

$$\bar{n}_q = \frac{1}{e^{\beta \omega_q} + 1},\tag{60}$$

with $\omega_q = -\cos q$, the details of the dispersion become irrelevant in the low-temperature limit, $\beta \to \infty$. As outlined in appendix B, the calculation of the corresponding slope $\bar{\gamma}_n(\beta, \lambda)$ can be carried out explicitly by linearizing the dispersion around the Fermi point and yields

$$\lim_{\beta \to \infty} \beta \, \bar{\gamma}_n(\beta, \lambda) = 2\pi \kappa_n(\lambda).\tag{61}$$

Hence, the slope ratios give exactly the expected CFT result in the limit $\beta \to \infty$.

Finally, one should emphasize that the key connection between the ground-state and quench scenarios is the eigenvalue relation (54), which is indeed identical for the two cases, even though it is satisfied only approximately for the quench with the initial state chosen here. It is easy to see that, by choosing instead a reflection symmetric Néel state with only even/odd sites occupied in the left/right half-chain, the second term in (52) vanishes identically. Hence one has $C'_{AB} = \lambda C_{AB}$ and using (49), the relation becomes exact even on the level of matrix elements. Furthermore, as shown in appendix C, the same holds true for the ground state of an initial Hamiltonian with a reflection symmetric staggered potential with arbitrary $\mu$. The derivation relies on a special property of the initial Hamiltonian that is completely analogous to (30), and the relation $C'_{AB} = \lambda C_{AB}$ holds exactly for finite $L$ and arbitrary times.

## 3.4 Entanglement revivals

For a chain of finite size, the result found in sec. 3.3 will be modified for times $t > L$, due to particles reflecting from the open boundaries of the chain. Such finite-size effects result in a sawtooth-like pattern of the entropy, as shown in Fig. 9 for the homogeneous case $\lambda = 1$. The

decay and revival of the entropy was studied for a periodic chain in [45], and the result can easily be generalized to the open chain by finding the proper contributions of the quasiparticle pairs after reflections. Indeed, for a pair with fixed velocity $v_q > 0$, the emission distance measured from the center of the chain must satisfy $x < \min(v_q t - 2L(n-1), 2L\, n - v_q t)$ in order to contribute after $n$ reflections from the boundary. Hence, the quasiparticle ansatz reads

$$S_n = \int_{-\pi}^{\pi} \frac{\mathrm{d}q}{2\pi} s_n(n_q) 2L \min\left[\left\{\frac{|v_q|t}{2L}\right\}, 1-\left\{\frac{|v_q|t}{2L}\right\}\right],\tag{62}$$

where the curly brackets denote the fractional part. The formula (62) is plotted with red solid lines in Fig. 9, showing a good agreement with the data, although the discrepancy increases for larger times due to subleading contributions.

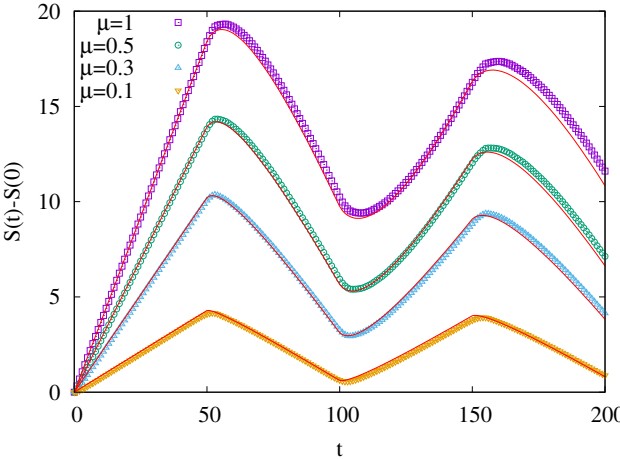

Figure 9: Entanglement decay and revival in the homogeneous quench for various $\mu$ and $L = 50$. The red solid lines show the quasiparticle ansatz (62).

The case of the defect requires a more careful analysis, by following the quasiparticle trajectories and scattering events for large times. This is shown on the left panel of Fig. 10, for a pair emitted at a distance $x$ from the defect. When reaching the defect, the transmitted part (solid line) of the red particle picks up an amplitude $s(q)$, whereas the reflected component (dashed line) receives an amplitude $r(q)$. For short times, the dominant contribution to the entanglement is created by the transmitted red and the blue particles. As discussed in sec. 3.3, there are some correlations between the transmitted and reflected components, however, their contribution to the entanglement is subleading for short times. Pictorially speaking, the dominant part of the entanglement is carried between the red and blue lines that reside in opposite halves of the system. For large enough times, however, the blue particle reaches the defect after a reflection and its transmitted part (solid line) can create entanglement with the reflected part (dashed line) of the red particle, and vice versa. These processes contribute in a time window depicted by the horizontal dotted lines, and both of them carry an amplitude $s(q)r(q)$, which we assume to add phase coherently. Finally, when the transmitted and reflected beams join again, they reproduce the original wave with unit amplitude, which is due to the fact that the amplitudes (34) do not carry any phase and satisfy $s^2(q) + r^2(q) = 1$.

The ansatz for the entropy can be found by simply summing up the contributions from the two different processes discussed above. The range of emission points $x$ of pairs that can contribute at a fixed time $t$ is illustrated on the top right panel of Fig. 10 for the first full period $0 < v_q t < 4L$ of time evolution. This has a triangular pattern, with the blue regions corresponding to the process that was already discussed in sec. 3.3, and described

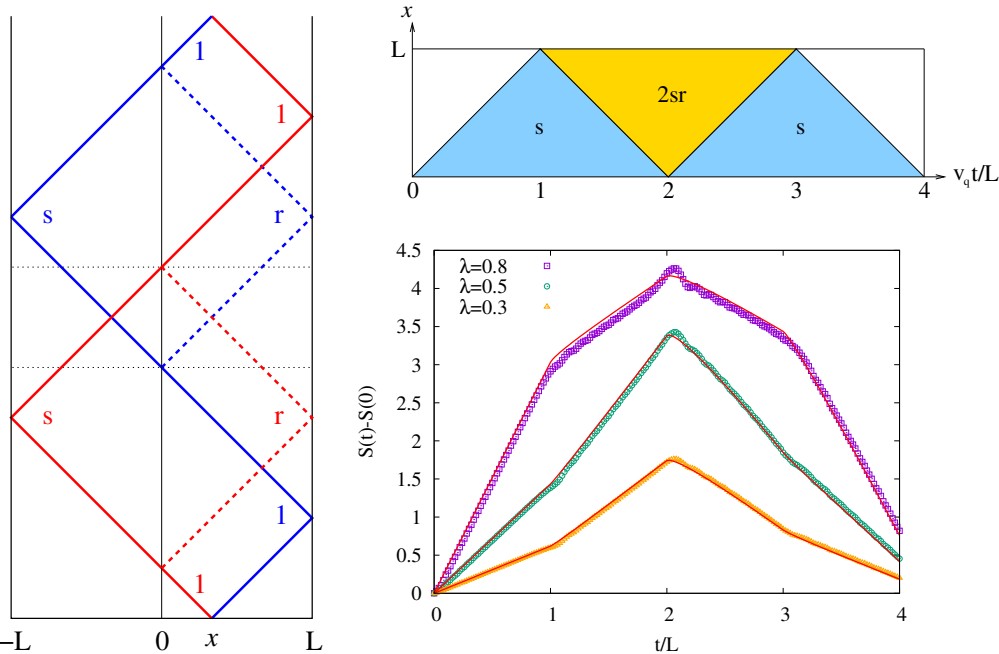

Figure 10: Left: quasiparticle trajectories with corresponding amplitudes. Top right: initial locations of quasiparticle pairs that contribute in the first full period $0 < v_q t < 4L$. The shaded regions show the contributions with the amplitudes indicated. Bottom right: entropy growth compared to the quasiparticle ansatz (64).

by the relation (55) for the occupation functions. Instead, in the yellow region one has an overall factor $2s(q)r(q)$ multiplying the correlations between the two halves of the chain, which suggests the analogous relation

$$\tilde{n}'_q(1 - \tilde{n}'_q) = 4\lambda^2(1 - \lambda^2) n_q(1 - n_q).$$ (63)

In other words, the entanglement carried between transmitted and reflected wave components is described via the modified occupation function $\tilde{n}'_q$. In turn, the quasiparticle ansatz reads

$$S_n = \int_{-\pi}^{\pi} \frac{dq}{2\pi} s_n(n'_q) 2L \min\left[\left\{\frac{|v_q|t}{2L}\right\}, 1 - \left\{\frac{|v_q|t}{2L}\right\}\right]$$
$$+ \int_{-\pi}^{\pi} \frac{dq}{2\pi} s_n(\tilde{n}'_q) L \max\left[0, 1 - \left|4\left\{\frac{|v_q|t}{4L}\right\} - 2\right|\right],$$ (64)

where the first and second terms follow from carrying out the integration over $x$ in the blue and yellow regions, respectively, taking into account the periodicity of the pattern.

The ansatz (64) is compared against the lattice data in the bottom right panel of Fig. 10, finding a remarkably good agreement. One should stress that the pattern of entanglement is completely different from the one found in Fig. 9. Indeed, instead of the decay in the time window $L < t < 2L$, one finds a continued growth of entanglement with a slope that can even exceed the one in the initial growth phase $0 < t < L$. The decay of the entropy ensues only after $t > 2L$, with a quasi-periodic pattern repeating itself after a full period $t = 4L$. The change in the behaviour is clearly due to the second type of contributions in (64). Note that for this result it is essential to consider the initial Hamiltonian (44) with a staggering that does not respect the reflection symmetry. Indeed, such an initial state yields a constructive interference for the correlations carried by the transmitted and reflected wave components, for pairs emitted on opposite sides of the defect. In sharp contrast, for reflection symmetric

staggering one obaints a destructive interference and, as shown in appendix C, the relation $C'_{AB} = \lambda C_{AB}$ holds for arbitrary times. In this case the second term in (64) is absent, and one obtains a completely similar pattern as in Fig. 9, which we verified numerically.

## 3.5 Hopping defect

To conclude this section, we remark that the quench results can also be generalized to other type of defects. We restrict our attention to short times $t < L$ and, for simplicity, we discuss the case of a hopping defect. This is given by a single weakened coupling $H'_{0,1} = H'_{1,0} = -\lambda/2$ in the center of the chain, without the local potentials. Setting $\lambda = e^{-\nu}$, the hopping defect is characterized by a transmission coefficient

$$T(q) = \frac{\sin^2 q}{\text{sh}^2 \nu + \sin^2 q}, \tag{65}$$

that depends on the momentum of the incoming particle, i.e. the defect is not any more conformal. In the quasiparticle interpretation, this suggests that the relation (55) between the occupation functions should be modified by replacing $\lambda^2 \to T(q)$, which yields

$$n'_q = \frac{1}{2}\left(1 \pm \sqrt{1 - T(q)\frac{\mu^2}{\cos^2 q + \mu^2}}\right). \tag{66}$$

Hence the slope of the quasiparticle ansatz (57) should be evaluated with the above occupation function. The result is compared against the numerics in Fig. 11 for $n = 1$, with a good agreement. Note that the main subleading term seems to be given by a constant for large times, without additional logarithmic contributions observed for the conformal defect.

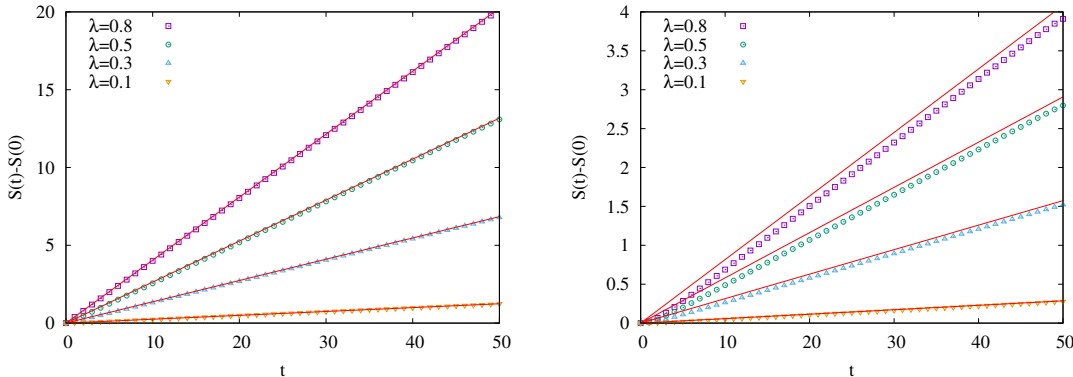

Figure 11: Entropy growth for the hopping defect after a global quench with $\mu \to \infty$ (left) and $\mu = 0.1$ (right), for various $\lambda$ and $L = 50$. The red solid lines show the quasiparticle ansatz with modified occupation function (66) given via the transmission coefficient (65).

## 4 Discussion

We studied the growth of entanglement after a global quench in the presence of a conformal defect, both on the level of CFT and for a free-fermion chain. In both cases, one finds a linear growth with a slope that is modified with respect to the homogeneous case. In the CFT context, introducing the concept of boundary twist fields and scaling dimensions, we could identify the

slope ratio with the effective central charge that governs the equilibrium scaling of the entropy, reproducing the results of [35]. On the lattice, however, we observed slight deviations of the slope ratios, found by a proper generalization of the quasiparticle picture. The origin of the discrepancy is that our choice of initial state, prepared by adding a staggered potential, does not perfectly reproduce a thermal distribution which is assumed in the CFT treatment. Indeed, the discrepancy vanishes by considering a thermal initial state in the low-temperature limit.

For short times, the decreased slope of the half-chain entropy can be attributed to the imperfect transmission across the defect. We also studied the large time behaviour on the lattice, observing an unusual pattern of entanglement revivals. These could be understood by following the quasiparticle trajectories after bouncing back from the end of the chain and adding the entanglement contribution carried between partially reflected and transmitted components. Altogether this creates a quasi-periodic pattern of the entropy. It should be stressed, however, that an important assumption is the phase-coherent superposition of the various contributions, which requires momentum-independent transmission and reflection amplitudes. Indeed, for the hopping defect this is not satisfied and one observes a highly irregular pattern of the entropy, which is probably due to interference effects and requires further investigation.

A possible future direction is the systematic investigation of the boundary twist fields introduced in this work. In particular, the determination of the boundary scaling dimension for generic CFTs is a nontrivial task. Furthermore, a complete characterization of the correlation functions of the boundary/bulk twist fields in a boundary geometry is fundamental to access the entanglement properties for more general subsystems. We hope to come back to this problem in the future, both for a better understanding of the entanglement in the ground-state [55–62] and in other non-equilibrium protocols, as the local quench [28, 35, 63]. In fact, in the latter case the entropy growth is logarithmic, and the prefactor is exactly given by $c_{\text{eff}}$ even on the lattice [28]. This is due to the fact, that in this low-energy quench protocol the precise form of the dispersion does not play a role.

Another interesting issue that we have left open is the question of logarithmic corrections beyond the quasiparticle ansatz. Similar corrections have been identified in the context of defect problems in [64, 65] for translational invariant steady states, and emerge due to a jump singularity in the occupation function. In our case, however, the logarithmic terms appear in the time domain and one has to deal with a non-translational invariant setting. Furthermore, some logarithmic corrections seem to appear even in the homogeneous quench setting (see Fig. 4), which must have some different origin and is left for future studies.

A further natural extension of our work is to study bosonic chains with defects. While the CFT derivation still applies, with an effective central charge obtained in [36], the quasiparticle ansatz is expected to be different from the fermionic case. One could also investigate other interacting models, such as the XXZ chain, where entanglement properties can be significantly influenced by the presence of defects [66–69]. It would be interesting to see whether the quasiparticle picture can be extended also for those systems.

## Acknowledgements

We would like to thank Pasquale Calabrese for discussions. VE acknowledges funding from the Austrian Science Fund (FWF) through Project No. P35434-N. LC acknowledges support from ERC under Consolidator grant number 771536 (NEMO).

# A  Diagonalization of the staggered chain

Here we show how to diagonalize the pre-quench Hamiltonian (44) for homogeneous hoppings ($\lambda = 1$), and calculate the occupation function of the post-quench Hamiltonian in the ground state of $\hat{H}_0$. We are interested in the thermodynamic limit, and thus assume periodic boundary conditions for simplicity. The Hamiltonian is then of size $2L$ and two-site translation invariant. Let us introduce the operators $a_m = c_{2m-1}$ and $b_m = c_{2m}$ and rewrite the Hamiltonian as

$$\hat{H}_0 = -\frac{1}{2}\sum_{m=1}^{L}(a_m^\dagger b_m + b_m^\dagger a_{m+1} + \text{h.c.}) - \mu\sum_m (a_m^\dagger a_m - b_m^\dagger b_m). \tag{A.1}$$

We first perform a Fourier transformation on both sublattices

$$\begin{pmatrix} a_m \\ b_m \end{pmatrix} = \frac{1}{\sqrt{L}}\sum_p e^{ipm}\begin{pmatrix} a_p \\ b_p \end{pmatrix}, \qquad p = \frac{2\pi}{L}k, \tag{A.2}$$

where $p$ is the sublattice momentum and its index runs over $k = -L/2+1, \ldots, L/2$. Introducing new operators via

$$a_p = \frac{1}{\sqrt{2}}\left(\sqrt{1 + \frac{\mu}{\Omega_p}}\,\alpha_p + \sqrt{1 - \frac{\mu}{\Omega_p}}\,e^{-i\frac{p}{2}}\beta_p\right), \tag{A.3}$$

$$b_p = \frac{1}{\sqrt{2}}\left(\sqrt{1 - \frac{\mu}{\Omega_p}}\,e^{i\frac{p}{2}}\alpha_p - \sqrt{1 + \frac{\mu}{\Omega_p}}\,\beta_p\right), \tag{A.4}$$

one arrives at the diagonal form of the Hamiltonian

$$H = -\sum_p \Omega_p(\alpha_p^\dagger \alpha_p - \beta_p^\dagger \beta_p), \qquad \Omega_p = \sqrt{\cos^2\frac{p}{2} + \mu^2}. \tag{A.5}$$

The half-filled ground state is given by all the $\alpha_p$ particles occupied $\langle \alpha_p^\dagger \alpha_p \rangle = 1$, whereas the $\beta_p$ band is empty, $\langle \beta_p^\dagger \beta_p \rangle = 0$.

The occupation numbers $n_q = \langle c_q^\dagger c_q \rangle$ for the homogeneous post-quench Hamiltonian are defined in terms of the Fourier modes on the full chain

$$c_q = \frac{1}{\sqrt{2L}}\sum_{j=1}^{2L} e^{-iqj}c_j, \quad q = \frac{2\pi}{2L}k, \tag{A.6}$$

where the index runs over $k = -L+1, \ldots, L$. One has then

$$n_q = \frac{1}{2L}\sum_{m,n=1}^{L}\left[e^{iq(2m-2n)}(\langle a_m^\dagger a_n \rangle + \langle b_m^\dagger b_n \rangle) + e^{iq(2m-2n-1)}\langle a_m^\dagger b_n \rangle + e^{iq(2m-2n+1)}\langle b_m^\dagger a_n \rangle\right]. \tag{A.7}$$

The ground-state expectation values can be evaluated using (A.2) and (A.4) as

$$\langle a_m^\dagger a_n \rangle = \frac{1}{L}\sum_p e^{-ip(m-n)}\frac{1}{2}\left(1 + \frac{\mu}{\Omega_p}\right), \quad \langle b_m^\dagger b_n \rangle = \frac{1}{L}\sum_p e^{-ip(m-n)}\frac{1}{2}\left(1 - \frac{\mu}{\Omega_p}\right), \tag{A.8}$$

as well as

$$\langle a_m^\dagger b_n \rangle = \frac{1}{L}\sum_p e^{-ip(m-n-1/2)}\frac{1}{2}\sqrt{1 - \frac{\mu^2}{\Omega_p^2}}, \quad \langle b_m^\dagger a_n \rangle = \frac{1}{L}\sum_p e^{-ip(m-n+1/2)}\frac{1}{2}\sqrt{1 - \frac{\mu^2}{\Omega_p^2}}. \tag{A.9}$$

Let us first consider the $q$ modes restricted to the reduced Brillouin zone $k = -L/2+1,\ldots,L/2$ of the sublattice. Substituting (A.8) and (A.9) into (A.7), the sums over $m$ and $n$ both give a factor $L\,\delta_{p,2q}$ and one arrives at

$$n_q = \frac{1}{2}\left(1 + \frac{|\cos q|}{\sqrt{\cos^2 q + \mu^2}}\right). \tag{A.10}$$

On the other hand, for $q$ values with $k = -L+1,\ldots,-L/2$ and $k = L/2+1,\ldots,L$ outside the reduced Brillouin zone, the $k$ index must be shifted to match the $p$ momentum, and the sums over $m$ and $n$ yield $L\,\delta_{p,2q\pm2\pi}$, respectively. The extra phase factors $\mathrm{e}^{\pm i(p/2-q)}$ appearing in the second and third term of (A.7) thus yield an extra sign and one has

$$n_q = \frac{1}{2}\left(1 - \frac{|\cos q|}{\sqrt{\cos^2 q + \mu^2}}\right). \tag{A.11}$$

However, in this regime the $\cos q$ function is negative and thus the two solutions (A.10) and (A.11) give an analytic function in the full Brillouin zone as reported in (46) in the main text.

## B  Calculation of the slope ratio for $\mu \to 0$

In this appendix we analyze the slope of the Rényi entropy (see Eq. (57))

$$\gamma_n(\mu,\lambda) = \int_{-\pi}^{\pi} \frac{dq}{2\pi}|\sin q|s_n\left(\frac{1}{2}\left(1 - \sqrt{1 - \lambda^2 \frac{\mu^2}{\cos^2 q + \mu^2}}\right)\right), \tag{B.1}$$

in the limit $\mu \to 0^+$, which requires a careful analysis. A preliminary observation is that for $\mu = 0$ one gets

$$\gamma_n(0,\lambda) = \int_{-\pi}^{\pi} \frac{dq}{2\pi}|\sin q|s_n(0) = 0, \tag{B.2}$$

which is a correct conclusion, since in this limit the pre/post-quench Hamiltonians are the same and there is no dynamics. However, here we are mostly interested in the way $\gamma_n(\mu,\lambda)$ goes to zero as $\mu \to 0$, to extract eventually the finite limit

$$\lim_{\mu\to 0}\frac{\gamma_n(\mu,\lambda)}{\gamma_n(\mu,1)}, \tag{B.3}$$

which we denote by $\frac{\gamma_n(0,\lambda)}{\gamma_n(0,1)}$ with a slight abuse of notation.

To proceed with the evaluation of (B.1), we first notice that the nontrivial contributions to the integral come from the values of the momentum $q$ close to $\pm\pi/2$, an observation which motivates the change of variable $q \to q + \pi/2$, and we get

$$\gamma_n(\mu,\lambda) = \int_{-\pi/2}^{\pi/2} \frac{dq}{\pi}\cos q \, s_n\left(\frac{1}{2}\left(1 - \sqrt{1 - \lambda^2 \frac{\mu^2}{\sin^2 q + \mu^2}}\right)\right). \tag{B.4}$$

At this point, it is natural to introduce the scaling variable

$$z \equiv \frac{\sin q}{\mu}, \tag{B.5}$$

so that we can write the slope as

$$\gamma_n(\mu,\lambda) = \mu \int_{-1/\mu}^{1/\mu} \frac{dz}{\pi} s_n \left( \frac{1}{2} \left( 1 - \sqrt{1 - \lambda^2 \frac{1}{1+z^2}} \right) \right). \tag{B.6}$$

Up to now, no approximation has been done and Eq. (B.6) is exact for any finite value of $\mu$. Nevertheless, the advantage of this expression comes from the fact that the singular behaviour of the integrand in Eq. (B.1) is not present anymore in Eq. (B.6), which makes the last expression suitable for a numerical evaluation.

From now on, we consider explicitly the limit of small $\mu$ and we extract only the term of order $O(\mu)$ in Eq. (B.6), obtaining

$$\frac{\gamma_n(\mu,\lambda)}{\mu} \simeq \int_{-\infty}^{\infty} \frac{dz}{\pi} s_n \left( \frac{1}{2} \left( 1 - \sqrt{1 - \lambda^2 \frac{1}{1+z^2}} \right) \right), \tag{B.7}$$

which is the main result of this appendix. While we were not able to perform analytically the integral (B.7) for any real value of $n$, and in particular for $n = 1$ directly related to the entanglement entropy, we provide below a simple closed expression for integer $n \geq 2$. The key observation is the following decomposition of the density of Rényi entropy

$$s_n(x) \equiv \frac{1}{1-n} \log(x^n + (1-x)^n) = \frac{1}{1-n} \sum_{p=-(n-1)/2}^{(n-1)/2} \log(e^{i2\pi p/n} x + (1-x)), \tag{B.8}$$

a simple identity which comes from the factorization of the polynomial $x^n + (1-x)^n$. We introduce for convenience the function $f(\alpha)$ as

$$\begin{aligned}
f(\alpha) &\equiv \frac{1}{2} \int_{-\infty}^{\infty} \frac{dz}{\pi} \log \left( \frac{1 - \sqrt{1 - \lambda^2 \frac{1}{1+z^2}}}{2} e^{i\alpha} + \frac{1 + \sqrt{1 - \lambda^2 \frac{1}{1+z^2}}}{2} \right) \\
&+ \frac{1}{2} \int_{-\infty}^{\infty} \frac{dz}{\pi} \log \left( \frac{1 - \sqrt{1 - \lambda^2 \frac{1}{1+z^2}}}{2} e^{-i\alpha} + \frac{1 + \sqrt{1 - \lambda^2 \frac{1}{1+z^2}}}{2} \right),
\end{aligned} \tag{B.9}$$

related to $\gamma_n(\mu,\lambda)$ by the following relation

$$\frac{\gamma_n(\mu,\lambda)}{\mu} = \frac{1}{1-n} \sum_{p=-(n-1)/2}^{(n-1)/2} f(2\pi p/n), \tag{B.10}$$

which is a straightforward consequence of Eq. (B.8). Interestingly, one is able to compute $f(\alpha)$ analytically, as we will show, and then a closed expression for the slope $\gamma_n(\mu,\lambda)$ for integer $n \geq 2$ can be provided. To do so, we first differentiate $f(\alpha)$ over $\alpha$ and we get

$$\frac{d}{d\alpha} f(\alpha) = - \int_{-\infty}^{\infty} \frac{dz}{\pi} \frac{\lambda^2 \sin(\alpha)}{2\lambda^2 \cos(\alpha) + 4z^2 - 2\lambda^2 + 4} = - \frac{\lambda^2 \sin(\alpha)}{2\sqrt{2\lambda^2 \cos(\alpha) - 2\lambda^2 + 4}}. \tag{B.11}$$

Then, we reintegrate back and obtain

$$f(\alpha) = \sqrt{1 - \lambda^2 \sin^2(\alpha/2)} - 1, \tag{B.12}$$

where the integration constant has been chosen so that $f(0) = 0$, a property which follows from the definition of $f(\alpha)$. Inserting (B.12) into (B.10) we finally arrive at the expression (59) reported in the main text.

We conclude this appendix with the computation of the slopes $\bar{\gamma}_n(\beta, \lambda)$ for an initial thermal state, highlighting its discrepancies with $\gamma_n(\mu, \lambda)$. The occupation number of such state at temperature $\beta^{-1}$ is

$$\bar{n}_q = \frac{1}{e^{-\beta \cos q} + 1}, \tag{B.13}$$

and the quasiparticle ansatz of the slope gives (see Eq. (57))

$$\bar{\gamma}_n(\beta, \lambda) = \int_{-\pi}^{\pi} \frac{dq}{2\pi} |\sin q| s_n \left( \frac{1 - \sqrt{1 - 4\lambda^2 \bar{n}_q (1 - \bar{n}_q)}}{2} \right). \tag{B.14}$$

While the latter expression is valid for any $\beta$, we are mostly interested in the low-temperature limit ($\beta \to \infty$), whose features are expected to be captured by CFT. After simple algebra, coming from the linearization of the dispersion around the Fermi-point $q = \pi/2$, one gets

$$\bar{\gamma}_n(\beta, \lambda) \simeq 2 \int_0^{\infty} \frac{dq}{\pi} s_n \left( \frac{1 - \sqrt{1 - \frac{\lambda^2}{\cosh^2 \frac{\beta q}{2}}}}{2} \right) = \frac{2}{\beta} \int_0^{\infty} \frac{dx}{\pi} s_n \left( \frac{1 - \sqrt{1 - \frac{\lambda^2}{\cosh^2 \frac{x}{2}}}}{2} \right), \quad \text{(B.15)}$$

a relation valid in the limit of large $\beta$. A tedious but straightforward calculation of the integral in Eq. (B.15), analogous to the computation of $\gamma_n(\mu, \lambda)$, gives for integer $n \geq 2$

$$\bar{\gamma}_n(\beta, \lambda) = \frac{1}{\beta} \frac{2\pi}{n-1} \sum_{p=-(n-1)/2}^{(n-1)/2} \left( \frac{1}{\pi} \arcsin \left( \lambda \sin \frac{\pi p}{n} \right) \right)^2. \tag{B.16}$$

If one compares the latter formula with the logarithmic prefactor in Eq. (43), valid for the ground-state entanglement (both in CFT and on the lattice), one realizes that they are proportional. This can be regarded as a non-trivial consistency check of the quasiparticle ansatz with the CFT results in the presence of a defect.

## C  Correlation matrix for reflection-symmetric staggered potential

In this appendix we consider a quench from the ground state of a chain with a conformal defect and an additional staggered field that is symmetric under reflection

$$\hat{H}_0 = \hat{H} + \mu \sum_{n=1}^{L} (-1)^n c_n^\dagger c_n - \mu \sum_{n=-L+1}^{0} (-1)^n c_n^\dagger c_n. \tag{C.1}$$

Note that the defect Hamiltonian $\hat{H}$ in (27) and thus the full Hamiltonian does not have a reflection symmetry. However, in the defect-free ($\lambda = 1$) case one does have a reflection symmetry, which is enough to show that the single-particle eigenvectors and eigenvalues of $\hat{H}_0$ possess the very same property (30) as those without the staggered field. Namely, one has

$$\tilde{\phi}_k'(m) = \begin{cases} \alpha_k \tilde{\phi}_k(m), & m \leq 0, \\ \beta_k \tilde{\phi}_k(m), & m \geq 1, \end{cases} \qquad \tilde{\Omega}_k' = \tilde{\Omega}_k, \tag{C.2}$$

with the coefficients given in (31). Note that we do not have a closed form expression for $\tilde{\phi}_k(m)$, but we will not need it. The only property we need is that, due to the reflection symmetry of $\hat{H}_0$ at $\lambda = 1$, the eigenvectors satisfy $\tilde{\phi}_k(1-m) = (-1)^{k-1} \tilde{\phi}_k(m)$.

We now calculate the time-dependent correlation matrix after the quench

$$C'(t) = e^{iH't} C'(0) e^{-iH't}, \tag{C.3}$$

where the post-quench Hamiltonian is the same as in the main text. Defining

$$A'_{k,l} = \sum_{i,j} \phi'_k(i) C'_{i,j}(0) \phi'_l(j), \tag{C.4}$$

one has for the matrix elements

$$C'_{m,n} = \sum_{k,l} \phi'_k(m) A'_{k,l} \phi'_l(n) e^{i(\omega'_k - \omega'_l)t}. \tag{C.5}$$

We are interested in the off-diagonal part ($m \leq 0$ and $n \geq 1$), where from (30) one obtains

$$C'_{m,n} = \sum_{k,l} \alpha_k \beta_l \phi_k(m) A'_{k,l} \phi_l(n) e^{i(\omega_k - \omega_l)t}. \tag{C.6}$$

Furthermore, one can rewrite $A'_{k,l}$ in terms of an overlap matrix

$$A'_{k,l} = \sum_{p \in F} B'_{k,p} B'_{l,p}, \qquad B'_{k,p} = \sum_{i=-L+1}^{L} \phi'_k(i) \tilde{\phi}'_p(i), \tag{C.7}$$

where the sum goes over the Fermi sea $F$, satisfying $\tilde{\Omega}_p < 0$. Using (C.2), the overlap matrix can be written as

$$B'_{k,p} = \alpha_k \alpha_p \sum_{i \leq 0} \phi_k(i) \tilde{\phi}_p(i) + \beta_k \beta_p \sum_{i \geq 1} \phi_k(i) \tilde{\phi}_p(i). \tag{C.8}$$

To proceed, let us first note the following identitites

$$\alpha_k \alpha_l = \begin{cases} \alpha_k^2, & \text{for } k-l \text{ even,} \\ \lambda, & \text{for } k-l \text{ odd,} \end{cases} \qquad \beta_k \beta_l = \begin{cases} \beta_k^2, & \text{for } k-l \text{ even,} \\ \lambda, & \text{for } k-l \text{ odd.} \end{cases} \tag{C.9}$$

Using the symmetry properties of both eigenvectors $\phi_k$ and $\tilde{\phi}_p$, one arrives at

$$B'_{k,p} = \left( \beta_k \beta_p + (-1)^{k+p} \alpha_k \alpha_p \right) \sum_{i \geq 1} \phi_k(i) \tilde{\phi}_p(i) = B_{k,p}, \tag{C.10}$$

which follows via (C.9), from the fact that the prefactor of the sum is nonvanishing only for even $k-p$, where it evaluates to $\alpha_k^2 + \beta_k^2 = 2$. The overlap matrix is thus completely independent of $\lambda$ and its matrix elements are nonvanishing only for $k-p$ even. This implies that $A'_{k,l} = A_{k,l}$ with nonvanishing elements for $k-l$ even. Finally, for $k-l$ even one has $\alpha_k \beta_l = \lambda$, and thus (C.6) yields the required identity $C'_{m,n} = \lambda C_{m,n}$ for the off-diagonal elements.

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
