# Peer review of "Entanglement evolution after a global quench across a conformal defect"

_SciPost Physics, doi:SciPost Phys. 14, 070 (2023)_

## Round 2 · Referee Report · Anonymous · 2022-10-15

Report
In the paper, the authors study entanglement dynamics after a global quench in 1D systems with conformal defects. Within the conformal field theory framework, the authors re-derive the relation between linear entanglement growth and the (effective) central charge of the system when including a defect. By comparing the CFT results to numerical simulations of lattice systems, analogies to quasiparticle pictures are drawn. Small discrepancies that arise between the CFT and lattice results are pointed out and discussed.
The paper is very well structured, clearly written, and discusses and interprets the scientifically sound results in detail. The analytical results both for the CFT and the lattice system are certainly very interesting and extensively compared. I recommend the paper for publication at SciPost, and list below only some minor remarks.
(i) As the authors show, there exist small differences between the exact CFT result and the lattice dynamics of the entanglement for a free fermion chain. It was mentioned that for local quenches these differences disappear (i.e., in the notation of reference [28], $\hat{c}_{eff} = c_{eff}$). Can the authors comment a bit more on the differences between local and global quenches in terms of the accuracy of the CFT result when compared to lattice systems?
(ii) By comparing the analytical/numerical results from the lattice system and CFT, it is stated that the deviations are a result from the initial state not being a perfect thermal description of the initial occupation. The authors use a gapped Hamiltonian with staggered field to initialize the state. Have the authors looked numerically at different ways to gap the Hamiltonian to get the initial state? If so, how do the results differ from their particular choice of initialization? In particular, is there a systematic way to approach a short-range entangled initial thermal state? By the end of Chapter 3.3, it is briefly mentioned that this discrepancy in fact vanishes when choosing a thermal initial state in the zero-T limit. Can the authors comment on this in a bit more detail?
(iii) When comparing the numerical results to Eq. (57), small deviations are seen. For small mass parameters mu, the authors find approximately constant corrections in time, whereas log-like deviations are observed for large mu. As they also appear for the homogeneous case (yellow curve in Fig. 4 / purple curve in Fig. 6), I assume this is not because of the correction terms of Eq. (53). Looking at the correction plots as a function of time in Figs. 4 and 6, it seems like there is some characteristic time that the lattice system takes to converge to the linear growth, where the slope at least for small mu seems to perfectly match the predictions. As even in the limit mu -> 0 the initial state does not correspond to a thermal state, I wonder what happens for the case for very small mu in terms of deviations to Eq. (57). As the discussion below Eq. (57) is quite short, a few more comments would be appreciated.
(iv) The results presented in Chapter 3.1 purely rely on analytical and numerical calculations of lattice systems. In order to compare with CFT, defects are chosen that are independent of wavenumber. In Chapter 3.5, the case of a hopping defect is discussed, which is no longer independent of the wavenumber. When choosing further different microscopic realizations of defects, say for instance the case of a resonant quantum dot (where there is an RG flow), have the authors thought about how entanglement growth can be characterized? Though this is certainly out of scope for this work, it would be nice to have some idea of what to expect in these type of settings.
Author: Luca Capizzi on 2022-11-10 [id 3005]
(in reply to Report 1 on 2022-10-15)
We thank the referee for positive judgment on our paper. We took into account the remarks, which are answered below. We believe the manuscript is now ready for publication without need of further revision.
(i) In order to compare the lattice results to the CFT predictions, it is necessary to understand the continuum limit of the microscopic system. For instance, while the agreement for the local quench is expected, as this protocol probes only the low-energy physics of the system, this is not the case for the global quench. Nevertheless, one could consider some specific low-energy limits, e.g. $\mu\rightarrow 0$, and try to compare them with field theoretic predictions. Still, our results suggest that the limit above may not exactly correspond to the field theoretic protocol considered in the manuscript.
We added a small clarification in the discussion: 'In fact, in the latter case the entropy growth is logarithmic, and the prefactor is exactly given by $c_{eff}$ even on the lattice [28]. This is due to the fact, that in this low-energy quench protocol the precise form of the dispersion does not play a role.'
(ii) We did not try to gap differently the initial Hamiltonian. A possibility could be the presence of a dimerized hopping, or the insertion of a pairing term, say $c_jc_{j+1}+h.c$.
However, we are not sure if there is a systematic way to design an initial Hamiltonian such that its ground state produces a given thermal occupation wrt the post-quench Hamiltonian.
The reason why the discrepancies between CFT and lattice vanish when considering a thermal initial state with $T\rightarrow 0$ is that the high-energy details of the lattice dispersion relation $\epsilon(k) = -\cos(k)$ become irrelevant in this limit. Thus, one can linearize the spectrum around the Fermi point and the CFT description emerges naturally.
(iii)
While for large times one expects a leading linear growth of the entropy, whose slope is investigated in our work, subleading corrections are expected, even in the absence of the defect. For the CFT, one expects that these corrections converge exponentially fast to a constant on a timescale of order $\beta$, as a more careful analysis of Eq. 23 shows. On the lattice side, however, it seems that an additional logarithmic correction is always present, even for small $\mu$. This is suggested by a closer observation of the corrections and by numerical calculations for larger sizes. We added an extra sentence below (57) to clarify this issue.
Unfortunately, however, we do not have a full understanding of these subleading terms.
(iv) We think that for free fermions the quasiparticle picture should be unaffected, no matter the microscopic details of the defect, and its features should be encoded in the single-particle reflection/transmission amplitudes which in general depend on a length-scale. The RG flow in the case of the resonant quantum dot is due to the vanishing reflection amplitude at the Fermi level. However, in the global quench one needs the full momentum-dependent transmission probability, as demonstrated for the hopping defect (Sec. 3.5).
Author: Luca Capizzi on 2022-11-24 [id 3066]
(in reply to Report 3 on 2022-11-20)We thank the referee for the positive judgement of our work. We answer below the questions
As the referee points out, it is not obvious why the slope ratio of the entropy growth should be related to the ground-state value for the lattice case. As we also mentioned in the text, it is quite remarkable that the two curves in Fig. 7 are very close to each other even in the $\mu\to\infty$ limit. On the other hand, we found it surprising, that the two curves do not match in the limit $\mu\to 0$, where one would naively expect the CFT result to hold. However, despite the fact that one has a low-energy quench in the latter case, the particular form of the occupation function still matters.
The presence of the defect introduces a mixing between the incoming modes of momentum $\pm k$. We speculate that for any pair $(+k,-k)$ one could associate a 2x2 reduced density matrix, and then a square root, coming from the diagonalization of the matrix, would appear naturally. However, we do not know how to formalize rigorously this idea.
Our numerical data suggest that the prefactor of the logarithmic corrections grows as $\mu$ grows. In the limit of $\mu \rightarrow \infty$, which gives the Neel initial state, we still observe these corrections. However, we do not fully understand their origin.
There are, in fact, some similarities with the case of dissipative defects, although the setup and the underlying physics is quite different. In the text, we'd rather refrain from a direct comparison with the dissipative case, as this is out of scope in the present context. We have, however, added these citations in the introduction, among the list of works dealing with free-fermion systems.

---

## Round 2 · Referee Report · Anonymous · 2022-11-4

Report
The authors study the entanglement entropy across a conformal defect separating two half-line CFTs , both at zero temperature equilibrium (ground state entanglement entropy) and after a global quench starting from some suitably chosen initial state.
First, they present some CFT predictions that the growth of the entanglement entropy after the quench should be linear in time, with an identical prefactor as that governing the ground state entanglement entropy, determined by the effective central charge associated with the defect. This reproduces results presented in another work, Ref [33].
Then the authors turn to a lattice free fermion calculation. While for the ground state entanglement agreement with the CFT predition is perfect, a discrepancy is found in the quench case. As explained by the authors, this is due to the choice of initial state : while the CFT treatment assumes a thermal distribution of quasiparticle speeds, the present computation uses as initial state the ground state of another gapped Hamiltonian, which should be derived by a non-thermal, "Generalized Gibbs" distribution. The authors then proceed to discussing the revivals occurring in systems of finite size.
This is a very clear and well-written paper, and the results are solid, with any discrepancy between the CFT and lattice approach well-commented and explained.
I recommend its publication in SciPost without the need for any major change.
Author: Luca Capizzi on 2022-11-10 [id 3004]
(in reply to Report 2 on 2022-11-04)We thank the referee for the positive judgment on our work.

---

## Round 2 · Referee Report · Anonymous · 2022-11-20

Strengths
1- very clear paper
2-in depth comparison between CFT and lattice results
Weaknesses
no major weaknesses
Report
In the paper the authors study the entanglement dynamics in the presence of a defect. They consider the situation in which there is a non trivial dynamics in the bulk of the chain, which is initially prepared in a low-entangled state that is not eigenstate of the Hamiltonian. They show that the entanglement entropy grows linearly with time. The authors derive an effective quasiparticle picture for the prefactor of the linear growth, which is compared with conformal field theory results (that they derive).
The results of the paper are interesting and address a timely topic. Moreover, the paper is quite well written and very easy to follow. My opinion is that this paper certainly deserves to be published in Scipost Phys.
I have some questions for the authors:
1) Why one should expect the ratio between the prefactors of the entropy growth should be the same as those of the ground state entropies? In the out of equilibrium situation that the authors explore this is not expected because the dynamics involves excited states and not just the low energy part of the spectrum. In fact I find already quite surprising that the two results are close (as shown in Fig. 7). It is not clear that even in the presence of conformal defect the relationship established in CFT should hold. Could the authors comment on that?
2) I find the result in formula (57) quite interesting. In fact this is different from the result in the case of a defect embedded in a Fermi sea.
In that case the entropy growth is still determined by the transmission coefficient of the defect but the formula has a much simpler interpretation. My question is whether is possible to provide any physical interpretation of (57). The square root structure of the formula suggests that perhaps it can be viewed as the entropy of a 2x2 reduced density matrix.
3) Finally, I have a question regarding Fig. 4. The authors find that there is a logarithmic correction to the quasiparticle picture at finite \mu. I was wondering whether the correction survives in the limit mu\to\infty.
4) Very recently there has been some interest in the entanglement evolution in the presence of dissipative defects. I think it would be interesting if the authors could compare their results with the ones in
SciPost Phys. 12, 011 (2022), arXiv:2209.14164.
They could also mention them in their work.
Requested changes
1) There is a typo: at the end of page 12 "offdiagonal" should be "off-diagonal"

---

## Round 4 · List of Changes

Added a sentence in the Discussion:
- ’In fact, in the latter case the entropy growth is logarithmic,
and the prefactor is exactly given by cef f even on the lattice [28]. This is due to the fact, that in this
low-energy quench protocol the precise form of the dispersion does not play a role'

-Slight modification of the sentence after Eq. 57:
'The subleading term for μ = 1 is likely to be logarithmic in time, with some superim-
posed oscillations, whereas it seems to be given by a constant for μ = 0.1. However, a closer
inspection of the latter case indicates that the corrections are probably still logarithmic,
albeit with a tiny prefactor, which is also supported by calculations for L = 200.'

---

## Editorial Decision

published